# Relationship between Behavioral Infant Speech Perception and Hearing Age for Children with Hearing Loss

**DOI:** 10.3390/jcm10194566

**Published:** 2021-09-30

**Authors:** Kristin M. Uhler, Alexander M. Kaizer, Kerry A. Walker, Phillip M. Gilley

**Affiliations:** 1Department of Physical Medicine and Rehabilitation, University of Colorado School of Medicine, Children’s Hospital Colorado, Aurora, CO 80045, USA; 2Department of Biostatics and Informatics, Colorado School of Public Health, University of Colorado Anschutz Medical Campus, Aurora, CO 80045, USA; alex.kaizer@cuanschutz.edu; 3Department of Otolaryngology-Head & Neck Surgery, University of Colorado School of Medicine, Aurora, CO 80045, USA; Kerry.walker@cuanschutz.edu; 4Institute of Cognitive Science, University of Colorado, Boulder, CO 80309, USA; gilley@colorado.edu

**Keywords:** infant speech perception, early intervention, congenital hearing loss, hearing aids

## Abstract

(1) Background: Research has demonstrated that early intervention for children who are hard-of-hearing (CHH) facilitates improved language development. Early speech perception abilities may impact CHH outcomes and guide future intervention. The objective of this study was to examine the use of a conditioned head turn (CHT) task as a measure of speech discrimination in CHH using a clinically feasible protocol. (2) Methods: Speech perception was assessed for a consonant and vowel contrast among 57 CHH and 70 children with normal hearing (CNH) aged 5–17 months using a CHT paradigm. (3) Results: Regardless of hearing status, 74% of CHH and 77% of CNH could discriminate /a-i/, and 55% of CHH and 56% of CNH could discriminate /ba-da/. Regression models revealed that both CHH and CNH performed better on /ba-da/ at 70 dBA compared to 50 dBA. Performance by hearing age showed no speech perception differences for CNH and children with mild hearing loss for either contrast. However, children with hearing losses ≥ 41 dB HL performed significantly poorer than CNH for /a-i/. (4) Conclusions: This study demonstrates the clinical feasibility of assessing early speech perception in infants with hearing loss and replicates previous findings of speech perception abilities among CHH and CNH.

## 1. Introduction 

This is a study about behavioral speech perception among hard-of-hearing infants whose hearing losses were identified through universal newborn hearing screenings. Infants in this study were fit with hearing aids by 3 months of age and participated in early intervention through a system of care called Early Hearing Detection and Intervention (EHDI) in the United States. Universal newborn hearing screenings and EHDI systems have radically improved age at identification, diagnosis, and early intervention for children who are hard-of-hearing (CHH), allowing speech perception testing at very young ages. While EHDI programs have led to significant improvements in language outcomes, much remains unknown about early speech perception abilities in young infants with hearing loss or how such abilities contribute to language outcomes compared to their normal hearing peers [1]. CHH who benefit from the EHDI programs demonstrate improved language, social-emotional, and academic skills [2,3,4] compared to children who are diagnosed and treated after two-years of age [2,5,6,7,8]. Further, early identification and treatment services have resulted in receptive and expressive language skills similar to children with normal hearing (CNH) [8,9,10]. Despite these improvements, there continues to be wide variability in spoken language outcomes (i.e., mean vocabulary quotient = 77.6, SD = 19.7; [11]), word learning abilities [12], and academic achievement [13,14,15] for CHH. Based on results from our previous research, we suggest that such variability in outcomes may be due to differences in speech perception abilities during infancy, which are poorer for infants with hearing loss [16], and may partially be explained by reduced auditory access to the full speech spectrum even after being fitted with hearing aids. 

Clinical assessment of speech perception is recommended as part of the Pediatric Minimum Speech Test Battery during the first year of life [17]; however, clinical tools are not readily available to objectively assess and validate hearing aid fittings and appropriateness of intervention strategies (e.g., hearing aid programming, transitioning to cochlear implants, or remote microphone technology). Performing “real-ear measures” is the current best practice for hearing aid verification in measuring the output of hearing aids in the outer ear canal. The sole utilization of hearing aid verification cannot ensure infants have access to the acoustic information needed to discriminate between speech sounds—a foundation for learning spoken language [6,14,18,19,20,21]. According to a survey of 117 pediatric audiologists, parent questionnaires remain the most frequently used clinical tool to assess speech perception for children under the age of two [22]; however, questionnaires are not objective measures of speech perception. Taken together, the lack of clinical tools and the variability of patient outcomes necessitates the translation of speech perception protocols from a research setting into clinical practice.

Numerous studies among CHH and CNH have examined speech perception abilities longitudinally [23,24,25,26]. However, we are interested in the first stage of perception as described by Aslin and Smith’s 1988 overview of the structural levels of perceptual development [27]. According to their model, perceptual development occurs in hierarchical stages consisting of three elements: (1) the “sensory primitives” or elementary perceptual units, (2) perceptual representations, and (3) higher order representations. Similar to the work of others (e.g., [25,28]) examining speech perception within this conceptual framework allows us to consider the first stage of development by examining an infant’s ability to detect a change in “sensory primitive” units of speech stimuli. A child’s inability to discriminate between two speech sounds could have a cascading effect on that child’s development of “perceptual representations,” and in turn, hinder the development of higher-order representations or the ability to form meaningful words/sentences from simple speech sounds. The inability of children to segment and differentiate speech stimuli from an ongoing stimulus would have deleterious impacts on language outcomes [29,30,31]. Therefore, it is essential to formulate a fundamental understanding of how infant speech perception develops among CHH compared to their CNH peers.

Among CNH, speech perception has been assessed in research settings for over 40 years and can be assessed behaviorally around 6 months of age using a conditioned head turn (CHT) paradigm. CHT is similar to a commonly used audiologic assessment tool, visual reinforcement audiometry (VRA). VRA is used to assess an infant’s ability to detect the presence/absence of sound and is clinically useful for establishing an infant’s audiometric thresholds [32]. Whereas CHT has been used to document a CNH’s ability to differentiate between vowels (/a-i/) and consonants (/ba-da/), translating this paradigm for clinical use requires systematic manipulation of test time and the number of test sessions while simultaneously maintaining scientific rigor of the test measure. For example, assessing speech perception in CNH has typically consisted of repeat testing if a child could not reach criterion on the speech perception task (e.g., [33,34,35,36]). This demonstrates that infants can improve their performance if they are seen multiple times for perceptual testing in a controlled research setting. CHT protocols have significant variability in attrition rates (10–15%; [37,38]) versus past studies which had attrition rates ranging from 5–50% [39] due to excluding data because infants became fussy, had abnormal middle ear function, or did not meet shaping/training criteria. Commonly, the numbers of children who did not meet shaping/training criteria are aggregated and their scores are not included in the statistical results. This approach to data aggregation leads to challenges in differentiating infants with poor speech discrimination abilities versus those who did not condition to the task or reach training criteria. Furthermore, the utilization of the shaping/training sessions add an additional study visit. For example, from 26% [40] to 33% of the participants’ data [35] has been excluded from previous reports and percentage of excluded participants does not encompass the number of infants who were excluded due to fussiness or ear infections. Taken together, these approaches result in more test sessions and smaller sample sizes. From a clinical perspective, if a child conditions to a task such as VRA, then clinicians may proceed with testing without further training and without detriment to the results. This concept is not novel and is commonly used in other speech perception tests where the child is taught the task such as the Early Speech Perception test [41] and the Open and Closed Set Speech Perception Test [42] using two to three practice words prior to test initiation. In the present study, we took a similar conditioning approach to Tsao and colleagues [30] here we conditioned to the task and then initiated testing once a child demonstrated task competency which is essential for future clinical utility. 

Assessment of speech perception among CNH has demonstrated that infants transition from being universal language learners and become better at discriminating their native languages between 6–12 months of age [43,44,45,46] as shaped by their language environment. Among CNH, speech perception abilities change over the first year of life as they become better at discriminating sounds in their native language. Infants also require a greater (louder) presentation level than adults to discriminate speech sounds [34,35,36,47] and the speech sound of interest must be louder than the background noise for infants to successfully differentiate between /ba/ and /da/. Nozza [36] assessed multiple intensity levels (50, 60, and 70 dBA) and compared performance for each intensity level to the softest sound both infants and adults could detect. For successful discrimination between the /ba/ and the /da/ the presentation level had to be 10–15 dB greater for infants compared to adults. These results suggest that utilizing a single intensity level as used for adults may result in poorer speech perception scores among infants [26,40,48,49]. 

While there is still more to learn about the development of speech perception among infants with normal hearing, even less is known about how development is impacted by the presence of hearing loss. Much of the work on speech perception development has been done in infants who use cochlear implants. Little is known about speech perception among infants with mild to severe permanent hearing loss between 6 to 17 months of age who benefited from early identification and hearing aid use. One finding is that CHH in the mild to severe hearing loss range discriminate vowel sounds more accurately than consonant sounds [26,37,38,48,50,51]. In our previous work, CHH (while wearing HAs) and CNH performed similarly on vowel discrimination (/a/ versus /i/). In contrast, only 50% of CHH and 71% of CNH discriminated /b/ and /d/ (Uhler et al. [37]). Among CHH only, vowel discrimination was better compared to consonant discrimination (*p* = 0.004), but the same was not true for the CNH (*p* = 0.45). 

The majority of speech perception studies for CHH have involved young deaf children who receive cochlear implants between 6–24 months of age and assessed speech perception between 9–30 months of age using CHT measures [26,52,53] visual habituation [24] or a modified visual habituation task [25]. Different speech stimuli have been employed, such as vowel stimuli [25,26,52], consonant contrasts [26,52], and suprasegmental differences (ahhh vs. hop hop) [54]. Broadly these findings suggest that following cochlear implantation, speech perception improves over time for young cochlear implant users. In contrast, Horn, Houston, and Miyamoto [23] found that following 1.4 months of cochlear implant use, 17-month-old listeners, were unable to discriminate “seepug” versus “boodup” through audition alone. When compared to their normal hearing peers, CHH may have different auditory development trajectories depending on various factors such as age at fit or degree of hearing loss. 

In this study, we extend our work towards a tool that could be clinically useful to assess speech discrimination in infancy [37,39]. A tool to objectively assess speech perception during infancy could help to determine if CHH are fitted appropriately and if they have access to acoustic cues allowing for speech discrimination (i.e., validation of hearing aid fittings). We set out to address three primary questions using a clinically feasible testing schedule: What proportion of CHH and CNH can demonstrate discrimination of each a vowel and consonant contrast as measured by CHT, at 50, 60, and/or 70 dBA SPL in a single test session while accounting for hearing age?Is there a difference in the ability of CHH (who met 1–3–6 benchmarks) and CNH in their demonstration of vowel and consonant discrimination assessed through a CHT task?Is there a relationship between aided SII measured at 50 dB SPL and performance on a CHT speech perception task for each contrast?

## 2. Materials and Methods

### 2.1. Participants

Data were collected for 129 children (58 CHH and 71 CNH) aged 5–17 months (mean (M = 9.61 months, standard deviation (SD) = 2.34 months. Two children (one CNH and one CHH) were lost to follow up prior to collecting speech perception data resulting in 127 children contributing to speech perception data (57 CHH, 70 CNH). 110 children completed speech perception testing for both consonant and vowel contrasts (/a/ vs. /i/ and /ba/ vs. /da/) the remaining 17 children completed either /a-i/ or /ba-da/. Specifically, nine CNH and two CHH only completed /a-i/ and three CNH and three CHH only completed the /ba-da/ contrast. The reasons for data loss were as follows: failure to condition to the task for the second contrast (three CNH, three CHH), loss to follow-up after completing one contrast (one CNH, one CHH), and modification in the presentation of the stimuli level for /ba-da/ 50 versus 65 dBA SPL for the first level assessed (eight CNH, one CHH). The data for 43 children with CNH [16] and 11 children with CHH [16] have also been reported in previous studies. Amplification and speech perception data were obtained from 57 infants (28 males and 29 females) with bilateral sensorineural hearing losses ranging from mild to severe. The better-ear pure tone average (500, 1000, 2000, and 4000 Hz) for the CHH group ranged from 18.33 to 83.33 dB HL (M = 41.3 dB, SD = 14.6 dB). All CHH wore bilateral air conduction hearing aids. The age at hearing aid fitting ranged from 1 to 6 months (M = 2.92 months; SD = 1.24 months), except for two children who were fit at 8 months of age. These two children were enrolled in early intervention, but their parents chose to delay fitting of hearing aids. For comparison purposes, data were obtained on 70 CNH (38 males and 32 females). The demographics for CHH and CNH appears in Table 1. 

Inclusion criteria were the same as in Uhler [41] (a) no evidence of significant developmental delays or secondary disabilities per parent report or as indicated in the electronic medical record, (b) demonstrated conditioned head turn in VRA, (c) normal tympanometry on the day of testing or patent pressure equalization tubes, (d) enrollment in early intervention, (e) use of hearing aids per parent report, and (f) either English or Spanish as the primary language spoken in the home. Additional inclusion criteria for the CNH were (g) parent report of infants passing their newborn hearing screening and the presence of otoacoustic emissions (OAEs) in both ears. Criteria for exclusion were (a) a history of untreated chronic middle ear infections paired with abnormal tympanometric findings on the day of testing and (b) auditory neuropathy.

### 2.2. Participant Hearing Aids

All participants used their own hearing aids during the CHT procedure. Following diagnosis, children received individualized care from their managing audiologist following best practices for amplification fitting, verification, and validation [55]. Children were fit with bilateral, behind-the-ear, air-conduction hearing aids coupled to custom earmolds with appropriate tubing, and filtered ear hooks. Hearing aids were programmed to prevent unnecessary signal distortion and managing audiologists confirmed that all participants’ hearing aids were programmed using Desired Sensation Level v5.0 (DSL) [56]. 

To verify amplification, simulated real-ear response measurements were compared to age-specific DSL targets using real-ear to coupler differences (RECDs). When RECDs were unable to be recorded from at least one ear from the child, average age specific RECD values were substituted. Before laboratory testing, all devices were evaluated to assure proper function. An electroacoustic test box measure was completed to assess hearing aid function. To assess hearing aid output either, measured or simulated real ear coupler differences. The aided SII was automatically calculated at 50, 60, and 70 dBA SPL using the Audioscan Verifit.

Hearing aid use was determined by reading the average daily hearing aid use from the child’s hearing aid. The mean hearing aid use was 6.62 h (SD = 4.08), see Table 2 for demographic characteristics unique to CHH. Each CHH had speech awareness thresholds and unaided pure-tones assessed using VRA. For the CNH, hearing was screened either using the same procedures as the CHH or utilization of otoacoustic immittance measures. 

Parents were paid $15 per hour for their child’s participation for each study visit.

### 2.3. Stimuli

For this experiment, two speech sound contrasts were used, /a-i/ and /ba-da/. Contrasts were selected based on difficulty levels, with vowels being the easiest and consonant contrasts being more difficult for both CHH and CNH [38,50,51,57]. Our natural speech tokens were produced by a female speaker, and adult listeners in the laboratory verified that the stimuli sounded natural. For procedures on how the stimuli were created, please refer to Uhler et al. [26]. All speech tokens were 500 ms in duration. For the /ba/ and /da/ stimuli, each consonant was 100 msec in duration, and the vowel duration was 400 msec. Stimuli were presented with a 1200 msec interstimulus interval during testing. All stimuli were equated for intensity via root-mean-square (RMS) amplitude normalization. Stimuli were presented at either 50, 60, and/or 70 dBA SPL. Sound pressure level of stimuli were measured in the sound field using an A-weighted scale and will be further referred to as dBA. Figure 1 shows a spectrogram of each stimulus to visualize formant differences.

### 2.4. Testing Protocol

All testing was conducted in an acoustically treated sound booth over two sessions. Session one included the case history (information related to the infant’s general health, development, and years of education of the infant’s mother), tympanometry, hearing screening, and, if time allowed, a threshold search for /a/ using CHT. Session two included a threshold search for /a/ (if not completed at the first visit) and the CHT assessment protocol. 

During CHT testing, one of the speech sounds for each stimulus pair, /a/ or /i/ and /ba/ or /da/, served as the background stimulus, repeated with 1200 ms interstimulus interval throughout the session. The other speech sound served as the target. The member of the pair serving as the target stimulus was counter-balanced across participants. The infant learned to respond by turning their head when the target stimulus was presented. 

The infant was accompanied by their caretaker into the sound booth for the CHT assessment. The infant was seated either on the caretaker’s lap or in a highchair in the center of the room to minimize distractions or task fatigue. Care was taken to ensure that the infant was comfortable and the location and distance to the speaker remained constant regardless of how the infant was seated. The background stimulus was on when the infant and caretaker entered the room. The speaker and visual reinforcement video screen were 90º to the right of the infant’s midline. An assistant who centered the infant’s gaze was positioned in front of the infant, slightly to the left. The caretaker and the assistant listened to music through supra-aural headphones to prevent them from hearing the sounds presented to the infant and inadvertently reinforcing the child or alerting the child to a contrast stimulus. 

The evaluator was seated outside the sound booth in a test room and observed the infant through a window. The evaluator in this study was one of five audiologists. The perception task consisted of two phases: conditioning and testing. During the conditioning phase, only change trials were presented so infants could learn to associate a change in the sound and the reinforcer. To facilitate learning during conditioning, the target sound was presented at 6 dBA louder than the background sound to draw the infant’s attention to the sound change. Initially, in conditioning trials, the reinforcer turned on following two target sounds to “teach” the task to the infant. The infant learned to pair a change in the sound with the reinforcer. After the infant made two consecutive head turns that occurred before the end of the first two presentations of the target sounds the intensity cue was removed. Once testing began, the evaluator could not hear the stimuli and was blinded to stimulus type—trials were initiated by the evaluator by pressing a button when the child’s attention was directed toward the midline.

Each of the 15 trials had an equal probability of being a change or no change trial; the trial type was randomly selected by the computer program. If the trial was a no-change trial, the background sound was presented three times. When a change trial was presented, the target sound was presented three times. Once the trial ended, the background sound continued. When head turns were observed by the evaluator, a button was pressed indicating the response. Correct responses were determined by the CHT software, and were rewarded by automatic presentation of an animated video. If the child’s head turn was incorrect, it was considered a false positive because no change had occurred. 

Fifteen trials were administered during each contrast assessment. Performance on speech perception was calculated using d-prime (d’) [58]. The advantage of d’ is that it eliminates the effect of response bias calculated from the number of false positives and hit rate. A “false positive” occurs when a child turned their head, but no change occurred in the stimuli (e.g., a-a-a). In this case, the button indicating that a head turn occurred would be pressed, but no reinforcement would occur. In contrast, a “hit” is when a child turns their head in response to a change in stimuli, which in turn would lead to the behavior being rewarded. False alarm rate is calculated by dividing the number of false alarms by total number of no change trials, and the hit rate is calculated by dividing the number of hits by the total number of change trials. D’ is calculated by using the *z*-score:
d’ = z(false alarms) − z(hits).(1)

If the child achieved a d’ value of at least 1.21 at 50 dBA, then testing was complete [34,36,38]. We hypothesized that if a child could successfully discriminate at a lower intensity level they would be able to do so at a higher intensity level [59]. For children who did not reach criterion at 50 dBA, the level was increased to 70 dBA, and testing resumed. Following completion of 15 trials at 70 dBA, regardless of performance, the presentation level was reduced to 60 dBA, and 15 trials were completed at that presentation level. Therefore, children who did not reach criterion at 50 dBA, a total of three conditions (/a-i/ at 50, 60, and 70 dBA) were completed, see Figure 2. In each session, testing continued until all conditions were completed or if the child was too fussy or tired to continue. On average, a single condition (i.e., /a-i/ at 50 dBA) was completed in five minutes and 32 s (SD = 5.35 min). 

### 2.5. Statistical Analysis 

Hearing age was calculated by subtracting age at hearing aid fitting (average age at hearing aid fitting was 3 months) from chronological age at test time. CNH hearing age was the same as chronological test age. Age was stratified as younger or older than nine months at time of testing, chosen as it matches the hearing age of CHH with the earliest age category of CNH that completed CHT testing (6–7 months) and better reflects an equal amount of auditory access between groups. Hearing status was categorized as normal hearing, mild hearing loss, and moderate or greater (moderate+) hearing loss. Hearing status and hearing age were analyzed as categorical variables instead of continuous variables given our intent to replicate previous literature with interpretable findings [11,60]. 

Descriptive summaries are provided as mean (standard deviation, SD) for continuous measures and as frequency (%) for categorical measures. Normality of the outcomes was evaluated with the Shapiro-Wilk test and graphically, with results suggesting normality. Comparisons of between group demographics used *t*-tests for continuous measures and chi-squared or Fisher’s exact tests for categorical measures, as appropriate. 

Linear regression models for the outcome of speech perception performance reported as d’ scores [58] were evaluated using generalized estimating equations with an exchangeable working correlation structure to account for individuals with multiple scores across different levels of intensity. Models adjusted for hearing age at testing, hearing status, and presentation level as predictors of speech perception performance. Among CHH, models were also fit to include aided SII at 50 dBA as a predictor of speech perception performance. 

Scatterplots with speech perception scores across presentation level and hearing status were created with regression fits based on models including PTA category (i.e., normal hearing 0–15 dB HL, mild 16–40 dB HL, and moderate+ ≥ 41 dB HL) and presentation level, with points jittered within a given intensity level (50, 60, 70 dB) to better visualize the data. All analyses and figures were conducted using R Foundation for Statistical Computing v3.6.3 (Vienna, Austria).

## 3. Results 

### 3.1. Effect of Hearing Status on Speech Perception 

Table 3 lists the proportion of infants who reached criterion (d’ ≥ 1.21) on each contrast as a function of presentation level and hearing status. Participants were included for the lowest presentation level at which they reached criterion and participants that did not reach criterion at any presentation level were placed in the “did not qualify” category for the respective contrast. Percentage totals were created using the total number of participants that completed testing for each respective contrast. For the vowel contrast, among CHH, 37% reached criterion on /a-i/ at 50 dBA and 74% reached criterion on /a-i/ at any presentation level. Similarly, among CNH, 31% reached criterion at 50 dBA and 77% reached criterion on /a-i/ at any presentation level. For the consonant contrast, both CHH and CNH groups performed similarly with 55% and 56% of each group, respectively reaching criterion at any presentation level. Therefore, most infants were able to discriminate between /a-i/ regardless of hearing status and only slightly more than half of the cohort could discriminate /ba-da/. Specifically, among CHH and CNH fewer children successfully discriminated /ba-da/ compared to /a-i/. 

### 3.2. Differences in Speech Perception Abilities among CHH and CNH

For analysis, we compared the average speech perception performance for both /a-i/ and /ba-da/ contrasts stratified by hearing age group (<9 months vs. >9 months), presentation level, and hearing status. There were no differences between CHH and CNH across conditions (*t*-test *p* > 0.05). 

### 3.3. Hearing Status

Next, we categorized speech perception performance as a function of hearing status (normal hearing, mild hearing loss, and moderate+ hearing loss) while considering hearing age. Performance on /a-i/ as a function of presentation level is shown in Figure 3a, and performance on /ba-da/ as a function of presentation level is shown in Figure 3b. Among CNH and children with mild hearing loss, the regression lines nearly overlap for performance on both contrasts. Children with moderate+ hearing losses demonstrated poorer performance for both the vowel and consonant contrasts than children with lesser degrees of hearing loss. For the /a-i/ contrast, there is a significant difference (W = −2.14, *p* = 0.03) such that children with moderate+ hearing losses had d’ scores that were 0.45 lower (95% CI (−0.86, −0.04)) than CNH. There were no group differences for /ba-da/ (*p* > 0.05 for all). 

### 3.4. Presentation Level

To examine the role of presentation level for vowel and consonant speech perception abilities, we fit linear regression models for the overall sample of CHH and CNH for the outcome of speech perception performance across all presentation levels (50, 60, and 70 dBA). For the overall sample in Table 4, both CHH and CNH performed better on /ba-da/ at 70 dBA compared to 50 dB presentation level and had d’ score that was 0.36 higher (95% CI (0.13, 0.58)) when adjusting for hearing age and hearing status (W = 3.09, *p* = 0.002). Among CHH cohort, Table 5, there is improvement in performance as a function of presentation level for /ba-da/ such that at 70 dBA compared to 50 dBA performance was 0.47 higher (95% CI (0.17, 0.77)) adjusting for hearing age and severity of hearing loss. No variables were significantly associated for /a-i/ for the cohort of CHH.

### 3.5. Aided SII

To examine the potential association of aided SII on speech perception abilities, the regression models were refit and restricted to only the CHH group. Aided SII at 50 dB input ranged from 0.21–0.98. Aided SII is not significantly associated with speech perception for /ba-da/ or /a-i/ when adjusting for age, mild versus moderate+ hearing loss, and presentation level (*p* > 0.05 for all).

## 4. Discussion

This study was designed to compare early speech perception abilities for CHH who met the EHDI guidelines and a group of CNH using a clinically feasible protocol. We examined whether performance varied as a function of contrast type (vowel and consonant), presentation level, hearing age, and hearing status. Both CHH and CNH were able to discriminate each contrast, however more children, regardless of hearing status, were able to reach criterion on the vowel contrast /a-i/ compared to the consonant contrast /ba-da/. CHH and CNH had higher scores on the /ba-da/ contrast at 70 dBA compared to 50 dBA; the same benefit was observed for the CHH when analyzed as a separate cohort. We also found that children with moderate+ hearing losses performed significantly poorer than CNH across multiple intensity levels for /a-i/. However, children with mild hearing loss did not differ in their performance from CNH for the contrasts tested in this study. Finally, aided SII was not a significant predictor of speech perception performance.

### 4.1. Hearing Status

Children with moderate+ hearing losses performed significantly poorer than CNH across multiple intensity levels for /a-i/ when considering hearing age. We were surprised that no differences in speech perception abilities were observed between CHH and CNH for /ba-da/ based on our previous research findings. Of note, all CHH benefited from meeting the EHDI benchmarks. Both groups, when considering chronological age, performed similarly, as 74% of CHH and 77% of CNH were able to successfully discriminate /a-i/ and 56% of CHH and 55% of CNH were able to discriminate /ba-da/. 

Forty-four percent of CHH and 45% CHH did not reach criterion for /ba-da/ at any level. Among CNH this is poorer than previously reported where 29% Uhler [37] and 28% Nozza [34] of 6–8-month-olds did not reach criterion on /ba-da/. Thus, this sample of CNH performed poorer on /ba-da/ whereas the CHH performed about the same as reported in Uhler et al. [37]. Overall, performance on /a-i/ was poorer compared to the performance reported in 2017; here 74% of CNH and 77% of CHH reached criterion at some presentation level. Previously, 85% and 95% respectively, of infants could discriminate /a-i/ [37]. It is possible that these differences in performance may be due to natural variability observed, due to an increase in sample size, and/or increasing the number of testers. In Uhler et al. [37,38] testing was done by a single tester whereas data presented in this paper was gathered by five different testers. 

When looking at our data for /ba-da/ discrimination, 56% of CHH and 55% of CNH reached a criterion, which is poorer than performance on /a-i/ discrimination. The finding that vowel discrimination is easier for infants, regardless of hearing status, is also similar to previously reported speech perception abilities found in the [26,37,38,48,49,61]. Overall, we have tripled our sample sizes from work reported in 2017 [37]. While the outcomes on speech perception of the vowel contrast remain similar for all infants and performance remains similar among CHH for /ba-da/, the same was not true for performance among CNH on the /ba-da/ contrast. Expressly, comparing results from Uhler et al. [37] to our present findings, performance among CNH declined from nearly 70% of 21 participants who were able to discriminate /ba-da/ at some presentation level to 56% among 70. These results suggest consideration of the duration of hearing aid use may be necessary when examining speech perception. Additionally, clinical use and implementation in a current manner is feasible for the /a-i/ contrast. We would expect CNH and children with mild hearing loss to perform similarly. If a child can discriminate /ba-da/ the clinical recommendations are relatively straight forward. However, if a child cannot discriminate the /ba-da/ contrast, clinical recommendations are limited, and further exploration is needed to understand these implications.

We speculate that better performance on /a-i/ than place of articulation contrasts such as /ba-da/ [26,37,48,49,61,62] may be due to the different roles that vowel and consonants play in language acquisition and learning [63] and that language specific vowels emerge before consonants [43,46]. These findings warrant further exploration on the perceptual development of vowels and consonants in CHH. We evaluated each contrast at multiple levels to determine if greater presentation levels improved performance, which we observed for /ba-da/. Replicating our previous work in a larger population of CHH who all benefited from meeting EHDI milestones, wore hearing aids, and had mild to severe hearing loss was an important first step towards expanding our work in the development of speech perception. Assessing infant speech perception would not have been possible before newborn hearing screening and the EHDI milestones. Our work supports that CHH can successfully discriminate these contrasts and that CHT is an efficient way to evaluate this behavior. However, future work in infant speech perception with CHH should be expanded to include more challenging phoneme contrasts (e.g., /s-sh/) [64], to examine performance in more realistic listening environments such as in background noise [35], and to explore the test-retest reliability in this population [55]. 

### 4.2. Presentation Level

CHH and CNH performed better on the /ba-da/ contrast at 70 compared to 50 dBA. The same benefit was observed for the CHH when analyzed as a separate cohort, while adjusting for age and severity. This difference in overall performance between 50 and 70 dBA agrees with the previous findings from Nozza [34] where infants performed optimally at 70 dBA. Nozza also found that 7/10 infants were not performing greater than chance and that testing at 50 dBA may not be reflective of any discrimination ability. These findings support utilization of a greater intensity (i.e., 70 dBA) level for /ba-da/ and may lead to improved performance on this contrast in a clinical setting. Further, assessing performance variability at different presentation levels could provide valuable information about how amplification algorithms such as noise cancelation, amplitude compression, or frequency compression might affect perceptual development in CHH. Future research to examine such effects could be useful for optimizing hearing aid fittings at different time points that coincide with developmental changes in perception.

### 4.3. Aided SII

As in our previous work, we examined the relationship between speech perception abilities and aided SII, because of the role it plays in hearing aid fittings [65,66]. Among CHH, aided SII is commonly used to determine proper amplification fitting based on an individual’s hearing loss. Our present results show that aided SII was not a significant predictor of speech perception for either contrast. These findings are partially inconsistent with our previous work [37] when we found that there was a relationship between speech perception abilities of /a-i/ and aided SII, but not /ba-da/ and aided SII. However, other previous studies have shown the aided SII may overpredict speech perception abilities in the pediatric population [64,65,66] and may not be a reliable indicator of predicted of speech perception performance. One explanation for these discrepancies is that aided SII is a measure derived from the expected intelligibility of acoustic speech properties in a typically functioning auditory system. However, the variability in etiologies and severity of hearing loss, coupled with the variability of amplification and treatment inherently lend to atypical acoustic, perceptual, and physiological characteristics that cannot be captured by such derived measures. These findings support inclusion of speech perception testing in addition to assessing aided SII in the clinic.

### 4.4. Limitations

There is a great deal of variability in performance of infant speech perception. We sought to test a large number of CHH and CNH, which we achieved. However, even when tripling our sample size of CHH, there is an uneven distribution of children with each degree of hearing loss ranging from mild to severe. The high variability can be seen from Figure 3a,b which depicts contrast performance across intensity levels and by hearing status. Variability is often a challenge when assessing this population as infants’ states change during and across test sessions due to factors such as fatigue, hunger, and interest [55]. Advancing infant speech perception work is challenging due to small sample sizes, high attrition rates, data inclusion/exclusion criteria, and high levels of variability. Cristia et al., [55] assessed test-retest reliability across three sites for varying ages (mean age range: 5.89–11.58 months) on 12 independent yet similar experiments each targeting some speech perception skill. In that study, the test-retest period ranged from 0–18 days within the first test date. Their findings revealed only three experiments with positive test-retest correlations, suggesting results from infant speech perception are highly variable. While we did not have multiple sites involved in testing, we did increase the number of clinicians administering the CHT assessment compared to our previous studies. Cristia et al.’s findings and close examination of our study highlight the need for replication and increased sample sizes in infant speech perception work. 

Furthermore, while we do not anticipate that CNH are unable to discriminate /ba-da/, our study design did not allow us to explore what it means that 44% of CNH were unable to reach criterion. However, this is the first work of its kind to report conditioning in a single test session and to report data for all infants who demonstrated successful conditioning to the task regardless of test performance. Future work should include some measure of test-retest reliability across multiple visits. Of note, a subset of our CNH was previously reported in Uhler et al. [16], for which they completed cortical evoked potential testing and exhibited passive discrimination of our consonant and vowel contrasts. The group tested in this study also lacks generalizability to CHH who did not meet the JCIH guidelines. Further research in this area could benefit from CHH that were either diagnosed with hearing loss and/or treated for hearing loss after the recommended EHDI timeline. However, withholding treatment for experimental purposes would not be ethical.

## 5. Conclusions

The majority of CHH (74%) and CNH (77%) were able to discriminate /a-i/ at intensity levels between 50–70 dBA in a single testing session. Reported data included all children who demonstrated successful conditioning regardless of their speech perception scores. However, there was a slight decline for CHH who were able to discriminate the /a-i/ contrast compared to our previous studies in which 95% of CHH could achieve criterion. When comparing discrimination performance of the consonant contrast, these findings are again similar among CHH such that 56% of CHH were able to discriminate /ba-da/ compared to our work published in 2017. The same pattern of performance did not hold true for CNH on the /ba-da/ contrast. Similar findings with a larger sample size contribute to the generalizability of our results among a population of CHH who met EHDI benchmarks. CHH can perform similarly to CNH in a quiet environment. This work contributes to the speech perception literature among infants with mild to severe permanent hearing loss and demonstrates the feasibility of using this clinically viable protocol.

## Figures and Tables

**Figure 1 jcm-10-04566-f001:**
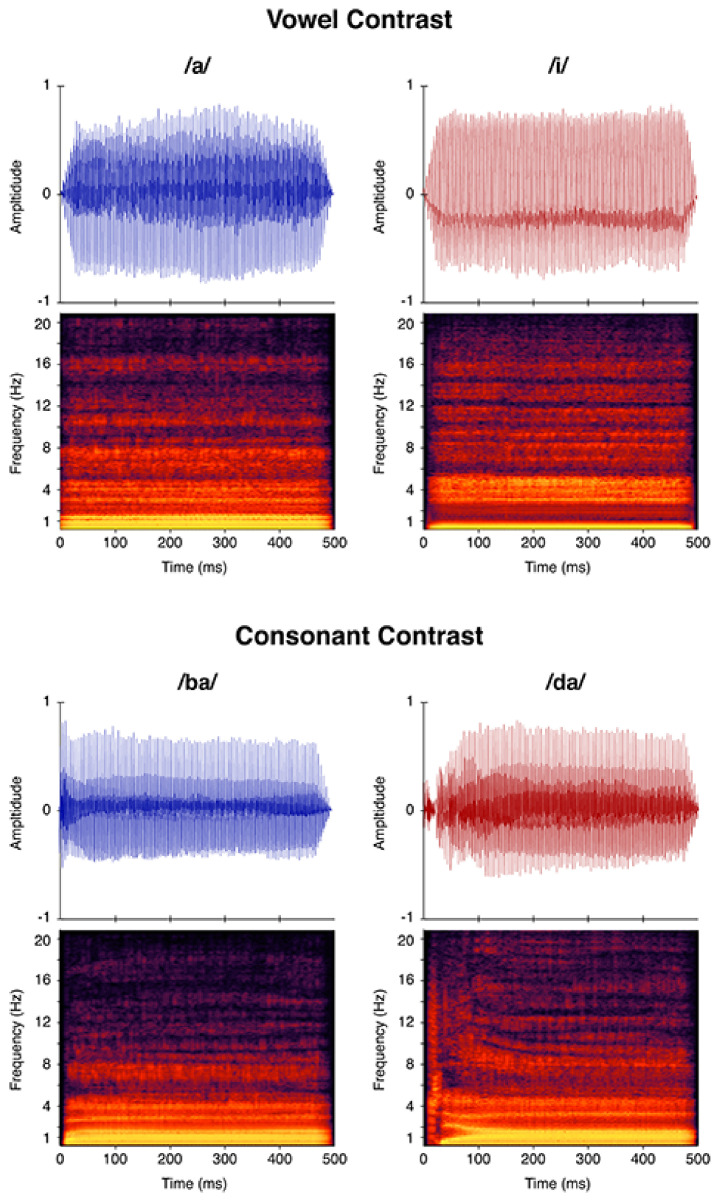
Time-amplitude waveforms and spectrograms for stimuli tested during the speech perception task. The top panel shows the /a/ and /i/ speech sounds for the vowel contrast and the bottom panel shows the /ba/ and /da/ speech sounds for the consonant.

**Figure 2 jcm-10-04566-f002:**
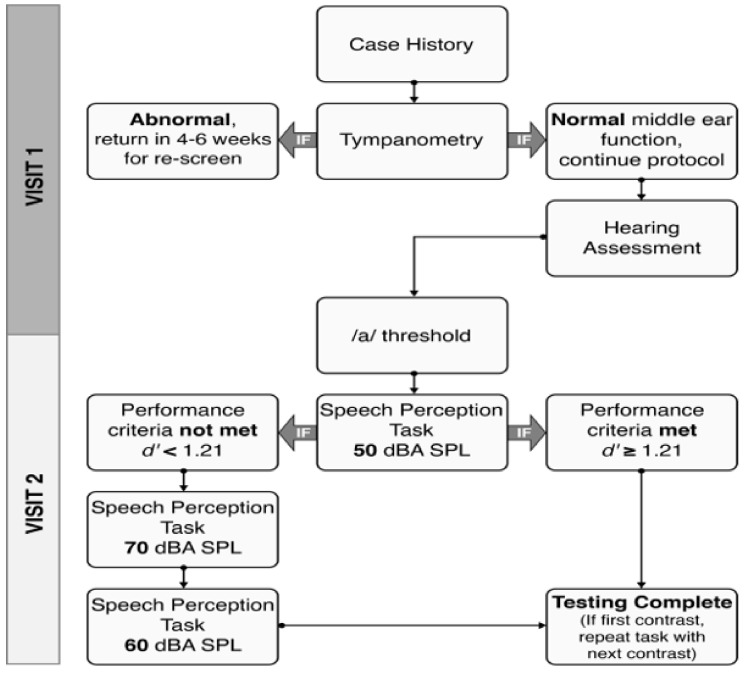
Flow-chart representing the testing protocol consisting of two visits: visit 1 is shown in the upper half of the chart and visit 2 in the lower half of the chart.

**Figure 3 jcm-10-04566-f003:**
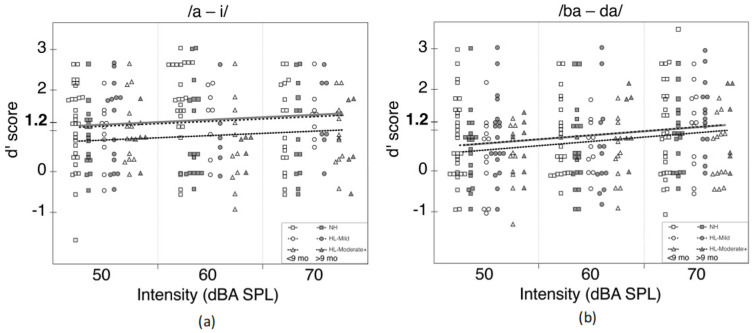
(**a**) Performance on /a-i/ as a function of intensity (50, 60, and 70 dBA SPL) represented in columnar format. Each intensity column contains d’ scores for each of three hearing groups: normal hearing (NH; squares), mild hearing loss (HL-mild; circles), and moderate+ hearing loss (HL-moderate+; triangles). Empty symbols represent participants with an age at test less than nine months and filled symbols represent participants with an age at test greater than nine months. Regression lines for each of the hearing groups as a function of intensity level are plotted across the intensity columns (thick grey line = NL; dashed line = HL-mild; dotted line = HL-moderate+). (**b**) Performance on /ba-da/ as a function of intensity (50, 60, and 70 dBA SPL) represented in columnar format. Each intensity column contains d’ scores for each of three hearing groups: normal hearing (NH; squares), mild hearing loss (HL-mild; circles), and moderate+ hearing loss (HL-moderate+; triangles). Empty symbols represent participants with an age at test less than nine months and filled symbols represent participants with an age at test greater than nine months. Regression lines for each of the hearing groups as a function of intensity level are plotted across the intensity columns (thick grey line = NL; dashed line = HL-mild; dotted line = HL-moderate+).

**Table 1 jcm-10-04566-t001:** Participant Characteristics with results presented as mean (standard deviation) for continuous measures and frequency (percent) for categorical measures.

Demographic	CHH	CNH	Statistical Test	*p*-Value
	(*N* = 57)	(*N* = 70)		
Male	28 (49.1%)	38 (54.3%)	Chi-squared	0.69
Age 9 months or Greater	30 (52.6%)	28 (40.0%)	Chi-squared	0.21
Hearing Age 9 months or Greater	14 (24.6%)	28 (40.0%)	Chi-squared	0.1
Age Category:			Fisher’s exact	0.006
6 to 7 months	7 (12.3%)	19 (27.1%)		
8 to 10 months	32 (56.1%)	42 (60.0%)		
11 to 13 months	13 (22.8%)	3 (4.3%)		
14 to 17 months	5 (8.8%)	5 (7.1%)		
Age < 5 months	0 (0.0%)	1 (1.4%)		
PTA Category:				
Normal (0–15 dB HL)	0 (0.0%)	70 (100.0%)		
Mild (16–40 dB HL)	39 (68.4%)	0 (0.0%)		
Moderate+ (≥1 dB HL)	18 (31.5%)	0 (0.0%)		
Hearing Age Category:			Fisher’s exact	<0.001
0 to 5 months	19 (33.3%)	1 (1.4%)		
6 to 7 months	22 (38.6%)	19 (27.1%)		
8 to 10 months	11 (19.3%)	42 (60.0%)		
11 to 13 months	5 (8.8%)	3 (4.3%)		
14 to 17 months	0 (0.0%)	5 (7.1%)		
PTA:	41.3 (14.6)	15.0 (0.0)		
Missing	2 (3.51%)	0 (0.0%)		
Age at CHT in months (M, SD):	10.2 (2.34)	9.16 (2.26)	*t*-test	0.02
Hearing Age in months (M, SD):	7.05 (2.49)	9.16 (2.26)	*t*-test	<0.001
Missing	0 (0.0%)	70 (100.0%)		

Note. This table summarizes participant characteristics for CNH and CHH including gender, mean chronological age at testing, mean hearing age at testing (Chronological age minus age at hearing aid fitting), threshold for /a/, and unaided pure tone audiometry (PTA), all children with hearing loss had bilateral permanent hearing loss.

**Table 2 jcm-10-04566-t002:** Demographics unique to CHH.

	Mean (SD/%)	Median (Min, Max)
Age at Hearing Aid Fit	2.92 months (1.24)	2.69 months (1.12, 8.03)
Datalogging	6.52 h (4.12)	6.55 h (0.00, 16.00)
Missing	9 (15.8%)	
Aided SII:		
50 dBA	0.73 (0.18)	0.74 (0.21, 0.98)
Missing	1 (1.75%)	

Note. Demographic variables for CHH, including age at hearing aid fitting, datalogging (average use per day), and aided SII for 50 dB input.

**Table 3 jcm-10-04566-t003:** Summary of speech perception.

		CHH			CNH		
Contrast	Level dBA SPL	Number of Participants	% of Participants	Mean Performance in d’ (S.D.)	Number of Participants	% of Participants	Mean Performance in d’ (S.D.)
/a-i/	50	20	37	1.94 (0.41)	21	31	1.87 (0.46)
	60	9	17	1.94 (0.46)	25	37	2.02 (0.54)
	70	11	20	1.66 (0.34)	6	9	1.80 (0.32)
Did not qualify		10	19		11	16	
Did not qualify, some missing		4	7		4	6	
Completed Testing		54			67		
Did not test		3			3		
/ba-da/	50	9	16	1.88 (0.61)	13	21	1.90 (0.62)
	60	10	18	1.77 (0.60)	8	13	1.98 (0.58)
	70	12	22	1.65 (0.31)	13	21	1.97 (0.63)
Did not qualify		20	36		25	41	
Did not qualify, incomplete testing		4	7		2	3	
Completed Testing		55			61		
Did not test		2			9		

Note: Data were included for the lowest level at which criterion was reached for the respective contrast. Due to rounding percentages to the nearest whole number, totals do not add up to 100%.

**Table 4 jcm-10-04566-t004:** Linear regression models for CHH and CNH.

Covariate	Estimate	95% CI	Wald Statistic (W)	*p* Value
Model 1-Outcome: d’ /ba-da/ for Overall Sample				
Intercept	0.28	(−0.32, 0.88)	0.90	0.37
Hearing Age at Test	0.04	(−0.02, 0.1)	1.33	0.18
Mild HL (vs. Normal)	0.07	(−0.24, 0.38)	0.46	0.65
Mod+ HL (vs. Normal)	−0.02	(−0.42, 0.38)	−0.08	0.94
60 Intensity (vs. 50 Intensity)	0.08	(−0.15, 0.30)	0.66	0.51
70 Intensity (vs. 50 Intensity)	0.36	(0.13, 0.58)	3.09	0.002 *
Model 2-Outcome: d’ /a-i/ for Overall Sample				
Intercept	1.31	(0.81, 1.80)	5.21	<0.001
Hearing Age at Test	−0.022	(−0.07, 0.02)	−0.97	0.33
Mild HL (vs. Normal)	−0.052	(−0.38, 0.28)	−0.31	0.76
Mod + HL (vs. Normal)	−0.45	(−0.86, −0.04)	−2.14	0.03 *
60 Intensity (vs. 50 Intensity)	0.149	(−0.09, 0.39)	1.21	0.23
70 Intensity (vs. 50 Intensity)	0.184	(−0.06, 0.43)	1.47	0.14

Note. Summary of data using generalized estimating equations (GEE) with an exchangeable working correlation structure to account for the multiple observations across level per individual. * denotes statistical significance.

**Table 5 jcm-10-04566-t005:** Linear regression models for CHH cohort only.

Covariate	Estimate	95% CI	Wald Statistic	*p* Value
Model 1HL-Outcome: d’ /ba-da/ for HL Only				
Intercept	−0.43	(−1.59, 0.73)	−0.73	0.47
Hearing Age at Test	0.03	(−0.06, 0.12)	0.61	0.55
Mod+ HL (vs. Mild HL)	0.15	(−0.30, 0.61)	0.66	0.51
60 Intensity (vs. 50 Intensity)	0.09	(−0.24, 0.43)	0.54	0.59
70 Intensity (vs. 50 Intensity)	0.47	(0.17, 0.77)	3.08	0.002 *
Aided SII at 50 dB	1.01	(−0.17, 2.19)	1.67	0.09
Model 2HL-Outcome: d’ /a-i/ for HL Only				
Intercept	1.15	(0.30, 1.99)	2.65	0.01
Hearing Age at Test	−0.05	(−0.11, 0.01)	−1.61	0.11
Mod+ HL (vs. Mild HL)	−0.24	(−0.66, 0.18)	−1.11	0.27
60 Intensity (vs. 50 Intensity)	−0.21	(−0.57, 0.16)	−1.10	0.27
70 Intensity (vs. 50 Intensity)	0.08	(−0.26, 0.42)	0.46	0.64
Aided SII at 50 dB	0.41	(−0.63, 1.45)	0.77	0.44

Note. Regression for CHH cohort only, using generalized estimating equations (GEE) with an exchangeable working correlation structure to account for the multiple observations across level per individual. * denotes statistical significance.

## Data Availability

De-identified data will be made available after institutional review and approval.

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
