# Peer review of "Relationship between Behavioral Infant Speech Perception and Hearing Age for Children with Hearing Loss"

_jcm, 2021, doi:10.3390/jcm10194566_

Round 1
Reviewer 1 Report
Abstract
- It is not clearly stated which was the objective of the study.
- The first sentence of the conclusion may not be necessary
Introduction
- The paragraph about the EHDI is essential?
Participants
- According to the 16-40 dB HL classification used, it is possible that some of the children presented a unilateral hearing loss?. PTA4 for the best ear ranged from 18.33 dB. Please clarify this.
Statistical Analysis
This section is hard to follow, with so many descriptions making it challenging to find the most relevant outcome. It should be improved to make it easier to read, otherwise the final message is blurred (as it is now).
Discussion
Most of the discussion refers to the description or comparison with previous studies. It will be beneficial if the authors address it in a more deeply manner.
Page 15, Line 430. “Performed better”, please clarify the idea. Fewer errors? higher scores?
Conclusions
The conclusion should be directly related to the study objectives. How it is written, just Line 472 it is a proper conclusion. Please rewrite it according to the goals.
Author Response
Dear Reviewers.
Thank you for your thoughtful and thorough review of our Manuscript ID: jcm-1385286,
Entitled: Relationship Between Behavioral Infant Speech Perception Abilities and
Hearing Age for Children with Hearing Loss. We have taken care to reduce similarities in our wording from from our previously published manuscripts, and addressed each of the reviewers suggestions in a point-by-point summary below. We feel these suggestions made by each reviewer has improved the readability and impact of this manuscript and hope you all agree.
Responses for Reviewer 1
Abstract
- It is not clearly stated which was the objective of the study.
Author response: We have added: The objective of this paper is to extend our work towards a tool that could be implemented clinically to assess infant speech perception.
- The first sentence of the conclusion may not be necessary
Author response: We have removed this sentence.
Introduction
- The paragraph about the EHDI is essential?
Author response: We have clarified, and the first paragraph now focuses on the essential components for our study.
Participants
- According to the 16-40 dB HL classification used, it is possible that some of the children presented a unilateral hearing loss?. PTA4 for the best ear ranged from 18.33 dB. Please clarify this.
Author Response: Thank you for your questions. Currently the section reads: Amplification and speech perception data were obtained from 57 infants (28 males and 29 females) with bilateral sensorineural hearing losses ranging from mild to severe. The better-ear pure tone average (500, 1000, 2000, and 4000 Hz) for the CHH group ranged from 18.33 to 83.33 dB HL (M = 41.3 dB, SD = 14.6 dB). All CHH wore bilateral air conduction hearing aids.
Furthermore, I have added this in the note: This table summarizes participant characteristics CNH and CHH including gender, mean chronological age at testing, mean hearing age at testing (Chronological age-age at hearing aid fitting), threshold for /a/, and unaided pure tone audiometry (PTA), all CHH had bilateral permanent hearing loss.
Statistical Analysis
This section is hard to follow, with so many descriptions making it challenging to find the most relevant outcome. It should be improved to make it easier to read, otherwise the final message is blurred (as it is now).
Author Response: We have edited for conciseness and provided an overview to explain our approach prior to introducing the statistical design. Hearing age was calculated by subtracting age at hearing aid fitting (average age at hearing aid fitting was 3 months) from chronological age at test time. CNH hearing age was the same as chronological test age. Age was stratified as younger or older than nine months at time of testing, chosen as it matches the hearing age of CHH with the earliest age category of CNH that completed CHT testing (6-7 Months) and better reflects an equal amount of auditory access between groups. Hearing status was categorized as normal hearing, mild hearing loss, and moderate or greater (moderate+) hearing loss. Hearing status and hearing age were analyzed as categorical variables instead of continuous variables given our intent to replicate previous literature with interpretable findings [11], [68].
Discussion
Most of the discussion refers to the description or comparison with previous studies. It will be beneficial if the authors address it in a more deeply manner.
Author response: We have focused the discussion and provided a summary within each paragraph about the potential clinical impact.
Page 15, Line 430. “Performed better”, please clarify the idea. Fewer errors? higher scores?
Author response: We have added: CHH and CNH had higher scores on the /ba-da/ contrast at 70 dBA compared to 50 dBA
Conclusions
The conclusion should be directly related to the study objectives. How it is written, just Line 472 it is a proper conclusion. Please rewrite it according to the goals.
Author response: Rewrote and focused conclusion to reflect goals.
Reviewer 2 Report
The authors wrote an article about Relationship Between Behavioral Infant Speech Perception Abilities and Hearing Age for Children with Hearing Loss.
The article is very interesting, well written with serious preparation about the topic. The article needs only few corrections: Please in the introduction talk more about cochlear implant surgery and its implication, using this article: Freni F, Gazia F, Slavutsky V, Scherdel EP, Nicenboim L, Posada R, Portelli D, Galletti B, Galletti F. Cochlear Implant Surgery: Endomeatal Approach versus Posterior Tympanotomy. Int J Environ Res Public Health. 2020 Jun 12;17(12):4187.
In the tables, please delete the column of the statistical test that you use and the columns with Wald Statistic (all is described in the statistical analysis yet).
In the discussion why you do not talk about a possibility to use real ear to a correct Hearing Aids utilization ? Gazia F, Galletti B, Portelli D, Alberti G, Freni F, Bruno R, Galletti F. Real ear measurement (REM) and auditory performances with open, tulip and double closed dome in patients using hearing aids. Eur Arch Otorhinolaryngol. 2020 May;277(5):1289-1295.
The conclusion is already reported in various articles in literature, so please emphasize the novelties of your studio.
Author Response
Dear Reviewers.
Thank you for your thoughtful and thorough review of our Manuscript ID: jcm-1385286,
Entitled: Relationship Between Behavioral Infant Speech Perception Abilities and
Hearing Age for Children with Hearing Loss. We have taken care to reduce similarities in our wording from from our previously published manuscripts, and addressed each of the reviewers suggestions in a point-by-point summary below. We feel these suggestions made by each reviewer has improved the readability and impact of this manuscript and hope you all agree.
Responses for Reviewer 2:
Comments and Suggestions for Authors
The authors wrote an article about Relationship Between Behavioral Infant Speech Perception Abilities and Hearing Age for Children with Hearing Loss.
The article is very interesting, well written with serious preparation about the topic. The article needs only few corrections: Please in the introduction talk more about cochlear implant surgery and its implication, using this article: Freni F, Gazia F, Slavutsky V, Scherdel EP, Nicenboim L, Posada R, Portelli D, Galletti B, Galletti F. Cochlear Implant Surgery: Endomeatal Approach versus Posterior Tympanotomy. Int J Environ Res Public Health. 2020 Jun 12;17(12):4187.
Author Response: This article you have suggested is an excellent article, thank you for your suggestion. However, this paper does not include any cochlear implant children. I have published a paper on VRISD in older children with Cochlear implants Uhler et al, 2011 and you are correct that this assessment can be used with cochlear implant users.
In the tables, please delete the column of the statistical test that you use and the columns with Wald Statistic (all is described in the statistical analysis yet).
Author Response: We must leave the statistical tests in because the only way it is replicable and for ease of review.
In the discussion why you do not talk about a possibility to use real ear to a correct Hearing Aids utilization ? Gazia F, Galletti B, Portelli D, Alberti G, Freni F, Bruno R, Galletti F. Real ear measurement (REM) and auditory performances with open, tulip and double closed dome in patients using hearing aids. Eur Arch Otorhinolaryngol. 2020 May;277(5):1289-1295.
Author Response: We could not agree more, that real ear measures are essential for the foundation of fitting hearing aids and documenting output of hearing aids at the level of the ear canal. However, it does not inform clinicians about discrimination abilities. We did complete real ear measures or simulations in all infants, see page 6. “To verify amplification, simulated real-ear response measurements were compared to age-specific DSL targets using real-ear to coupler differences (RECDs). When RECDs were unable to be recorded from at least one ear from the child, average age specific RECD values were substituted. Before laboratory testing, all devices were evaluated to assure proper function. An electroacoustic test box measure was completed to assess hearing aid function. To assess HA output either, measured or simulated real ear coupler differences. The aided SII was automatically calculated at 50, 60, and 70 dBA SPL using the Audioscan Verifit.”
Furthermore, all infants in this paper used fully occluding earmolds and no one used hearing aids coupled with domes.
The conclusion is already reported in various articles in literature, so please emphasize the novelties of your studio.
Author Response: We have refocused our conclusion and removed the first sentence as you suggested and modified the paragraph, I think your suggestions have markedly improved the flow and interpretability.
Comments and Suggestions for Authors
The authors wrote an article about Relationship Between Behavioral Infant Speech Perception Abilities and Hearing Age for Children with Hearing Loss.
The article is very interesting, well written with serious preparation about the topic. The article needs only few corrections: Please in the introduction talk more about cochlear implant surgery and its implication, using this article: Freni F, Gazia F, Slavutsky V, Scherdel EP, Nicenboim L, Posada R, Portelli D, Galletti B, Galletti F. Cochlear Implant Surgery: Endomeatal Approach versus Posterior Tympanotomy. Int J Environ Res Public Health. 2020 Jun 12;17(12):4187.
Author Response: This is an excellent article, thank you for your suggestion. However, this paper does not include any cochlear implant children. I have published a paper on VRISD in older children with Cochlear implants Uhler et al, 2011 and you are correct that this assessment can be used with cochlear implant users.
In the tables, please delete the column of the statistical test that you use and the columns with Wald Statistic (all is described in the statistical analysis yet).
Author Response: We must leave the statistical tests in because the only way it is replicable and for ease of review.
In the discussion why you do not talk about a possibility to use real ear to a correct Hearing Aids utilization ? Gazia F, Galletti B, Portelli D, Alberti G, Freni F, Bruno R, Galletti F. Real ear measurement (REM) and auditory performances with open, tulip and double closed dome in patients using hearing aids. Eur Arch Otorhinolaryngol. 2020 May;277(5):1289-1295.
Author Response: We could not agree more, that real ear measures are essential for the foundation of fitting hearing aids and documenting output of hearing aids at the level of the ear canal. However, it does not inform clinicians about discrimination abilities. We did complete real ear measures or simulations in all infants, see page 6. “To verify amplification, simulated real-ear response measurements were compared to age-specific DSL targets using real-ear to coupler differences (RECDs). When RECDs were unable to be recorded from at least one ear from the child, average age specific RECD values were substituted. Before laboratory testing, all devices were evaluated to assure proper function. An electroacoustic test box measure was completed to assess hearing aid function. To assess HA output either, measured or simulated real ear coupler differences. The aided SII was automatically calculated at 50, 60, and 70 dBA SPL using the Audioscan Verifit.”
Furthermore, all infants in this paper used fully occluding earmolds and no one used hearing aids coupled with domes.
The conclusion is already reported in various articles in literature, so please emphasize the novelties of your studio.
Author Response: We have refocused our conclusion and removed the first sentence as you suggested and modified the paragraph, I think your suggestions have markedly improved the flow and interpreteability.
Round 2
Reviewer 1 Report
Changes ok